

# Exploring the interactions among SNARC effect, finger counting direction and embodied cognition

Giulia Prete and Luca Tommasi

Department of Psychological, Health and Territorial Sciences, "G. d'Annunzio" University of Chieti-Pescara, Chieti, Italy

## ABSTRACT

The Spatial Numerical Association of Response Code (SNARC) is the preferential association between smaller/larger magnitudes and left/right side, respectively. Some evidence suggest a link between SNARC and a left-to-right finger counting habit. We asked 268 participants to show how they use the hands to count from 1 to 10. By means of this ecological task, 80% of the sample use first the right hand (to count from 1 to 5) and the majority of them use a palm-up posture. In Experiment 2 ($N = 46$) right-starters were asked to categorize 1-to-5 magnitudes as even or odd, using the left and right hand. Stimuli were presented both as Arabic numbers and by means of left and right hand photographs in palm-up and palm-down posture. Results confirmed the expected SNARC effect in the Arabic condition. With hand images we found that right hand responses were better for larger than for smaller magnitudes (SNARC, mainly for left hand palm-up stimuli), showing that the SNARC can be generalized to different codes. Finally, the interactions between magnitudes and left/right hand images in palm-up and palm-down posture suggest that embodied cognition can influence numerical processing.

## INTRODUCTION

### The Spatial-Numerical Association of Response Codes (SNARC)

The association between numbers and space has been widely shown in decades of research (for a recent review see *Guida & Campitelli, 2019*). For instance, in Western culture, smaller/larger numbers are categorized faster with the left/right hand, respectively. This association between small-left and large-right is also evident in magnitude irrelevant tasks, such as parity judgments (i.e., categorizing numbers as even or odd). This automatic association is known as Spatial-Numerical Association of Response Codes (SNARC) effect, and it was firstly described by *Dehaene, Bossini & Giraux (1993)*. In this seminal study, participants were asked to categorize one-digit numbers (range: from 0 to 9) as even or odd: the results revealed that participants were faster at categorizing smaller numbers with the left hand and larger numbers with the right hand. The authors explained the SNARC effect as due to a Mental Number Line (MNL), a mental representation of quantities,

Corresponding author
Giulia Prete, giulia.prete@unich.it

organized from the leftmost to the rightmost portion of an imaginary horizontal line, in which smaller elements are positioned towards the leftmost pole and larger elements are positioned towards the rightmost pole. According to this model, the detection of a magnitude leads to the automatic placement of such a number size in a specific position along the MNL, and to the following facilitation in responding to it by using the ipsilateral hand (e.g., left hand for "leftward placed" magnitudes). *Dehaene, Bossini & Giraux (1993)* also showed that the same number is preferentially associated with a left or right response depending on whether it is larger or smaller within a specific range, meaning that the magnitude of a number is relative to that of the specific set of numbers used (e.g., 4 and 5 are small in a 4–9 range, but they are large in a 0–5 range). This pioneering study also revealed that the SNARC is a consistent and pervasive effect, being present in both right handed and left handed individuals. However, the effect is weak in individuals educated in a right-to-left writing system (see also *Zebian, 2005*), and thus the authors concluded that it is dependent upon a cultural and educational substrate. The strength of the effect has been further confirmed in different tasks: for instance, *Fischer (2001)* found that when participants are asked to localize the midpoint of digit strings, the bisection is mislocalized either to the left or to the right with respect to the real center of the strings, according to the magnitude of the digits used (e.g., leftward/rightward for "11111" and "99999", respectively). Similar results were found by using number words of the same length but referred to smaller or larger numbers (*Calabria & Rossetti, 2005*), revealing the independence of the effect from the specific code used (e.g., Arabic numbers, letters, strings). More recent findings, however, suggest that culture or education do not have a causal role in the development of the SNARC effect, because it has been found that a similar effect is present also in non-human species (e.g., *Rugani et al., 2015*; *Vallortigara, 2018*), as well as in newborns (*Di Giorgio et al., 2019*).

The existence of an automatic SNARC effect is now unquestionable, even if it is still not clear whether it is mainly due to cultural reasons, to finger counting habits, or it is an innate bias.

## Different hypotheses on the origin of the SNARC: Mental Number Line vs. finger counting direction

The most acknowledged explanation of the SNARC is the shared parietal substrate of numbers (magnitude) and space (*Walsh, 2003*), corresponding to the neural basis for the MNL. In support of this view, it is well-known that the parietal lobe is a region crucial for visuospatial processing, and that a parietal damage can cause the Gerstmann syndrome, involving dyscalculia, agraphia, finger agnosia, and left–right confusion (*Gerstmann, 1940*). Moreover, a functional near-infrared spectroscopy study confirmed that the hemodynamic signature of the SNARC effect is in the bilateral intraparietal sulcus and in the left angular gyrus (*Cutini et al., 2014*).

An alternative hypothesis has been proposed for the SNARC effect, based on the direction of the finger counting habit: the "manumerical hypothesis" (*Fischer & Brugger, 2011*) posits that the association between magnitude and space develops during childhood, and that the cerebral areas involved in finger and number processing overlap, at least partially

(e.g., *Thompson et al., 2004*). According to this hypothesis, the SNARC effect in Western culture would be due to the fact that finger counting often starts with the left hand, so that smaller numbers (from 1 to 5) are located on the left (left hand), and larger numbers (from 6 to 10) are located on the right (right hand). The hypothesis is supported by *Fischer (2008)*, who found that the SNARC effect was strong in participants who used the left hand to start counting (left-starters), and it was absent in participants who used the right hand to start counting (right-starters). In particular, the author carried out two tasks: Experiment 1 was aimed to designate participants as left- or right-starters, by means of a printed questionnaire containing a schematic drawing of two supine, palm-up hands (thumbs pointing outwards). Participants were invited to imagine counting with their fingers from 1 to 10 and then to write the corresponding number next to each finger on the drawing. Fischer found that, in a sample of Scottish participants, 66% were left-starters and only 34% were right-starters, and that the result was independent of participants' handedness. In Experiment 2, carried out by a subgroup of participants, the author confirmed the expected SNARC in a group of left-starters, but no reliable SNARC was found in a group of right-starters. The higher proportion of left-starters, with respect to right-starters, was then replicated by *Lindemann, Alipour & Fischer (2011)*, who asked participants to hold out their empty hands in a supine position before finger counting, and then to remember this image and to complete a computerized version of the test, selecting the matching numbers for each finger shown on a computer screen. Less than the 10% of participants showed a different performance between the paper-and-pencil and the computerized version of the test, and the authors interpreted this result as a confirmation of the validity of the large scale paper-and-pencil test used by *Fischer (2008)*. It has to be highlighted that in the same study the authors also revealed the importance of the culture on finger counting habits, with 68% of left-starters in Western participants and 63.4% of right-starters in Middle-Eastern participants. Of note, among Western participants, Italians and Belgians did not reveal a significant preference for starting to count with the left hand, differently from participants from the Netherlands, United Kingdom, Canada, Finland, Germany and the United States (*Lindemann, Alipour & Fischer, 2011*). However, *Lucidi & Thevenot (2014)* found that 26% of a sample of adults revealed a mismatch between a questionnaire and an ecological task on finger counting. Similarly, *Wasner et al. (2014)* confirmed that finger counting is influenced by the strategy of counting used: they found that 28% of participants started counting with their left hand in a spontaneous counting condition, but this percentage increased to 54% in a left-to-right perceptual arrangement of fingers, and it grew up to 62% in a left-to-right perceptual arrangement condition in which the participant' dominant hand was busy. More recently, *Cipora et al.* (*2019*; Supplementary materials) found a higher proportion of left- than right-starters in Russian and Polish participants (Slavic language), but a higher proportion of right- than left-starters in participants speaking Germanic (German, English), Uralic (Hungarian), Oghuz (Turkish), Romantic (Spanish, Italian, Romanian) languages, as well as in Iranian (Farsi) participants, whose reading/writing direction is from right to left. Thus, the authors revealed the effect of culture on the counting direction, with a focus on the language families. *Sato & Lalain (2008)* found that in a sample of 117 French participants of different age groups, 69% were right-starters

and just 26% were left-starters. *Di Luca et al. (2006)* tested an Italian sample by using an original task: the authors asked participants to identify Arabic digits by pressing one of ten keys by using all of the ten fingers. They found that a right-to-left hand-digit mapping led to better performance than did a left-to-right hand-digit mapping. Moreover, the authors also found that such a mapping was faster than the SNARC-congruent mapping both when participants carried out the task in palm-up and in palm-down postures (*Di Luca et al., 2006*). A right-to-left hand counting habit has been recently confirmed in a Polish sample by *Hohol et al. (2018)*, who exploited a modified version of the Fischer's questionnaire: in particular, the authors firstly asked participants to count with their fingers from 1 to 10 and to memorize the order, and only in a following moment they were presented with schematic hands drawings on which they were asked to mark the respective numbers. Moreover, in a parity judgment task during Transcranial Magnetic Stimulation (TMS), an increase in amplitude of motor-evoked potentials of right-hand muscles was found during the presentation of smaller rather than larger numbers (*Sato et al., 2007*). This result is in accordance with other brain imaging evidence, showing a key role of the precentral gyrus (together with the parietal areas) in both spatial encoding and number processing (*Dehaene et al., 1999*; *Dehaene et al., 1996*; *De Jong et al., 1996*; *Pesenti et al., 2000*; *Pinel et al., 1999*; *Rueckert et al., 1996*).

We can conclude that the two main theories about the origin of the SNARC are (i) that suggesting a central role of the MNL (*Dehaene, Bossini & Giraux, 1993*), according to which we spontaneously place smaller/larger magnitudes in the left/right side of the space, respectively, and (ii) the ''manumerical hypothesis'' (*Fischer, 2008*), according to which the placement of smaller/larger magnitudes in a left-to-right axis would be associated to the habit to start finger counting from the left hand (smaller numbers) and to proceed through the right hand (larger numbers). It has been shown that in all cases the cerebral areas involved in the SNARC are the same areas activated during spatial tasks, but the specific role of culture in its development is still unclear.

## Exploring the SNARC in an embodied cognition framework

The findings reviewed above lead to study the SNARC effect and the finger counting in a body-related point of view. According to the embodied cognition models, cognition, body and environment are considered as parts of an integrated system (e.g., *Mahon & Caramazza, 2008*). In support of this frame, *Di Luca & Pesenti (2008)* showed that participants named numerical finger configurations faster when they were conform to the finger configurations usually used by the participants than when they did not. Importantly, they also found a facilitation in a magnitude judgment task, when an Arabic target stimulus was primed by an unconsciously perceived numeral finger configuration, mainly if the prime showed a canonical configuration of finger counting. The authors concluded that canonical finger counting configurations can activate number semantics in an automatic fashion. In this view, *Atmaca et al. (2008)* showed a joint SNARC effect when two participants performed a parity tasks together. Moreover, by changing the position of the responding keyboard, *You et al. (2018)* found that the object-based reference frame is

not related to the SNARC effect, but that the position of the body of two participants tested together (egocentric coding) influenced the task.

*Riello & Rusconi (2011)* asked participants whose typical counting routine proceeded from thumb-to-little finger on both hands to perform a magnitude task and a parity task, by using the index finger and the middle finger of either the left or the right hand, keeping their hands palm-down or palm-up. The authors found a unimanual SNARC effect with each hand, specifically for participants responding with the right hand in the palm-down posture and with the left hand in the palm-up posture. The evidence has been ascribed to a posture-invariant structural representation of the hands, which seems to be stronger than the pure effect of the left-to-right MNL. Different results were found in a cross-modal task: *Brozzoli et al. (2008)* presented one digit number on a computer screen (visual presentation) and asked participants to detect a tactile stimulus applied either on the thumb or on the little finger. The authors found that when the participants' right hand was in a palm-down posture, the performance was better when tactile stimuli were delivered to the little finger after the presentation of number 5 than number 1. However, when the participants' right hand was in a palm-up posture and tactile stimuli were delivered to the little finger, the performance was better after the presentation of number 1 than number 5. They concluded that in case of competition between space- and body-based representation of numbers, the human brain "prefers" the former, showing the stronger role played by the MNL in spatial cognition. Similarly, it has been shown that the SNARC disappears in the crossed hands condition (*Wood, Nuerk & Willmes, 2006*), suggesting that it is dependent upon the mental and/or physical disposition of the body in the space. A more complex pattern of results was found by *Viarouge, Hubbard & Dehaene (2014)*, who found the expected SNARC in both uncrossed and crossed hands conditions, but only when instructions were focused on the left/right response buttons; the effect disappeared in the uncrossed condition when instructions were focused on the responding hand, as well as when the responses were given on the vertical axis (top/bottom buttons), concluding that the SNARC effect is dependent upon a number of spatial reference frames, and it can be modulated by the specific experimental context.

## SNARC effect, embodied cognition and finger counting

Starting from the contrasting results found in different studies, revealing—in left-to-right writing systems—a population bias in starting the finger counting with the left hand (e.g., *Fischer, 2008*; *Lindemann, Alipour & Fischer, 2011*) or with the right hand (e.g., *Sato & Lalain, 2008*; *Hohol et al., 2018*), the first aim of the present study was to assess this bias in a wide Italian sample. Moreover, due to the controversial results about the possible correspondence between paper-and-pencil, computerized and ecological tasks, we asked participants to physically show how they use the fingers to count from 1 to 10 (Experiment 1; ecological task). Furthermore, we randomly selected a subgroup of these participants and administered to them a classic parity judgment task, in which they were invited to use the left or right hand to categorize even or odd numbers presented on a computer screen (Experiment 2). The second aim of the present study, in fact, was to confirm the SNARC effect assessed by means of Arabic numeral stimuli (from now on Arabic code)

by using a restricted set of magnitudes (1-to-5), and to verify the possible existence of the SNARC effect (i.e., better performance with the left/right hand for smaller/larger numbers) in a sample that starts finger counting with the right hand. Specifically, Experiment 2 was mainly aimed at disentangling the origin of the SNARC effect: according to the MNL (*Dehaene, Bossini & Giraux, 1993*), the SNARC effect should be independent from the direction of finger counting and it should be based upon a spontaneous placement of lower/higher magnitudes on the left/right side of the space, respectively; according to the manumerical hypothesis (*Fischer & Brugger, 2011*), the SNARC effect should be due to the habit of starting counting with the left hand. Thus, both of these two theories predict that participants who start finger counting from the left hand would show a classic SNARC effect in a parity judgment task, but they differ in the prediction on right-starters: participants who start finger counting from their right hand would show the SNARC effect, if the MNL is the basis of the effect, but they would not reveal the SNARC effect if the manumerical hypothesis is true. Considering these premises, only right-starters were involved in Experiment 2, with the aim to verify which of the two main hypotheses on the origin of the SNARC is valid. Starting from the literature about SNARC and embodied cognition (e.g., *Riello & Rusconi, 2011*; *Sato et al., 2007*), and from the evidence that hand configurations allow the automatic activation of number semantics (*Di Luca & Pesenti, 2008*), we also built an original condition in which the same magnitudes used in the classic parity judgment task were presented, but instead of the Arabic code we presented numbers as left/right hand photographs during finger counting (e.g., number five showed by means of an open hand with the five fingers stretched, in an ecological perspective). Images were kept as natural as possible (e.g., without removing shadows) in order to make the computerized task as much "ecological" as possible. For both the Arabic code and the photographed stimuli, we asked participants to carry out the same parity judgment task, expecting to find a similar SNARC effect. Finally, starting from the different results found depending on the postures of the participants' hands (palm-up, palm-down; e.g., *Di Luca et al., 2006*; *Riello & Rusconi, 2011*), instead of manipulating participants' hand posture, we manipulated the postures of the stimuli (left and right hands were shown either palm-up or palm-down). We expected that, according to the SNARC effect, for both Arabic code and hand images the performance of participants would be better for smaller/larger numbers categorized with the left/right hand, respectively (expecting to find a significant interaction between magnitude and responding hand). We also hypothesized that, according to an embodied perspective, the photographs of a left/right hand would lead to a better performance with the corresponding left/right hand used to respond (this could be revealed by the interaction between the depicted hand used as stimulus and the responding hand). Finally, we also aimed at exploring whether this hand-compatibility effect could be stronger for stimuli presented palm-down (as this was the posture of the participants' hands during the task) or for stimuli presented palm-up (as the viewed hands during a first-person finger counting). Thus our hypotheses were: (i) a better performance with the left/right hand for smaller/larger magnitudes (SNARC effect); (ii) a stronger SNARC effect when magnitudes are shown by means of left/right hand images (embodied cognition); (iii) a stronger SNARC by means of left/right hand mainly when stimuli are

presented palm-down (compatible with the hand posture held by participants during the task)".

# EXPERIMENT 1
## Material and methods
### Participants and procedure
Participants were contacted by consulting a list of university students created for research purposes. Only right-handed volunteers were contacted and the task was carried out by a final sample of 268 healthy participants (132 female and 136 male) with an age comprised between 18 and 39 years old (mean ± standard error: 24.4 ± 0.01 years). Participants were tested in isolation and the task required less than 5 min.

The present study did not involve patients, children or animals, as well as drugs, genetic samples or invasive techniques, thus it was not subject to ethical review by the academic medical research board. Nevertheless, verbal informed consent was obtained from all participants and the experiment was conducted in accordance with the ethical standards prescribed by the Declaration of Helsinki.

Participants were invited to enter a silent and empty room, taking care that they did not have something in the hands, and to stand in front of the experimenter. The experimenter asked the participant to show the way in which they usually used the fingers to count from 1 to 10. No reference was made to the hand to be used and no feedback was given to the participant. The experimenter observed the finger counting made by each participant and categorized the "type of action". Finger counting was categorized regarding: (1) the hand used to count from 1 to 5 and from 6 to 10 (left or right); (2) the position of the hands during the counting (palm-up or palm-down); (3) the order of the fingers "used" in each hand (e.g., from little finger to thumb or from thumb to little finger).

After the finger counting, which was observed by the experimenter, participants were invited to take part in different studies, after which a computerized version of the handedness inventory was administered and then they were debriefed. The handedness score was measured by means of the Italian version of the Edinburgh Handedness Inventory (*Salmaso & Longoni, 1985*), in which a score of -100 corresponds to a complete left preference, and +100 corresponds to a complete right preference (handedness score: 71.25 ± 0.06).

## Results
Results are summarized in Table 1: a total of 205 participants (76.49%) started the finger counting with the right hand (from 1 to 5) and then used the left hand (from 6 to 10); 56 participants (20.89%) started the finger counting with the left hand (from 1 to 5) and then used the right hand (from 6 to 10); the remaining 7 participants (2.61%) used the right hand to count from 1 to 5 and then re-used the same hand to count from 6 to 10. A chi-square test showed that the difference in starting hand was significant ($X^2_{(1)} = 86.92$, $p < 0.001$), with the majority of the sample starting the finger counting by using the right hand.

**Table 1 Pattern of finger counting from 1 to 10 in Experiment 1: frequencies (column N) and percentages (column %).** The first column indicates the hand posture: UP for "palm-up", DW for "palm-down". Letter pairs represent the hand (L for "left"; R for "right") and the finger used to indicate the number at the top of the column (T, thumb; I, index; M: middle; R: ring; P, pinkie).

| | 1 | 2 | 3 | 4 | 5 | 6 | 7 | 8 | 9 | 10 | N | % |
|---|---|---|---|---|---|---|---|---|---|---|---|---|
| **UP** | LT | LI | LM | LR | LP | RT | RI | RM | RR | RP | 52 | 19.40 |
| **DW** | LT | LI | LM | LR | LP | RT | RI | RM | RR | RP | 1 | 0.373 |
| **UP** | LT | LI | LM | LR | LP | RP | RR | RM | RI | RT | 3 | 1.119 |
| **UP** | LT | LI | LM | LR | LP | LT | LI | LM | LR | LP | 0 | 0 |
| **UP** | RT | RI | RM | RR | RP | LT | LI | LM | LR | LP | 200 | 74.63 |
| **DW** | RT | RI | RM | RR | RP | LT | LI | LM | LR | LP | 3 | 1.119 |
| **UP** | RT | RI | RM | RR | RP | LP | LR | LM | LI | LT | 2 | 0.746 |
| **UP** | RT | RI | RM | RR | RP | RT | RI | RM | RR | RP | 7 | 2.612 |
| | | | | | | | | | | | 268 | 100 |

Moreover, two chi-square tests were carried out to compare the orientation of the hands within the subgroups of left- and right-starters: the significant result showed that finger counting was made preferentially in palm-up position in both the left-starters subgroup (palm-up: 52 vs palm-down: 1; $X^2_{(1)} = 49.07$, $p < 0.001$) and the right-starters subgroup (palm-up: 200 vs palm-down: 3; $X^2_{(1)} = 191.18$, $p < 0.001$). Note that all participants assumed either a palm-up or a palm-down posture, without "ambiguous" postures (e.g., hands perpendicular to the ground).

In order to assess the possible effect of the handedness score on the finger counting habits, the sample was divided into 3 subgroups considering the score obtained in the handedness questionnaire. Due to the fact that all participants were right-handed and that the possible handedness score was comprised between 0 (no handedness preference) and 100 (total right preference), we considered a first subsample of participants who scored less than 33.33 ($28.81 \pm 1.26$; Group 1: no preference, $N = 5$), a second subsample for scores comprised between 33.34 and 66.66 ($56.04 \pm 0.88$; Group 2: weak right preference, $N = 95$), and the last subsample for scores higher that 66.67 ($81.12 \pm 0.71$; Group 3: strong right preference, $N = 168$). For each subgroup a chi-square test was carried out comparing the frequency of left-starters vs right-starters. After Bonferroni correction for multiple comparisons, the result in Group 1 was not significant, possibly due to the very low sample size of this subsample (2 vs 3, $X^2_{(1)} = 0.02$), but it was significant in Group 2 (21 vs 67, $X^2_{(1)} = 20.90$, $p < 0.001$) and in Group 3 (33 vs 135, $X^2_{(1)} = 59.50$, $p < 0.001$), confirming a higher proportion of right-starters in both weak and strong right preference groups.

## Discussion

In a sample of 268 right-handed participants we found that 79.1% starts finger counting with the right hand and 20.9% starts finger counting with the left hand. The difference is significant, showing that in the Italian population there is a bias to start counting with the right hand. This result is in contrast with the results described by Fischer in a Scottish sample (66% left-starters vs 34% right-starters), but it has to be noticed that in that study a pen-and-pencil questionnaire was used. We can hypothesize that the left-to-right

writing direction could have had an influence on the results by Fischer, even if contrasting evidence has been recently collected in samples from different countries by *Cipora et al. (2019)*, suggesting that culture has a key role in influencing finger counting direction. Our results are in line with those collected in samples geographically closer to the one we tested, such as Germans (28% left-starters; *Wasner et al., 2014*) and French (26% left-starters, *Sato & Lalain, 2008*), supporting a central role of culture on finger counting. Finally, we also found that finger counting is not influenced by the degree of hand preference, due to the fact that both weak and strong right hand preference groups revealed a significant prevalence of right-starters. This latter result could be viewed as in line with the evidence that hand preference is not related to finger counting (*Fischer, 2008*) even if caution is needed in this regard, due to the fact that only right-handed participants were tested in the present study.

## EXPERIMENT 2

Experiment 2 was aimed at verifying the second main hypothesis of the present study, namely the possibility that even in a sample of right-starters (as the majority of the sample tested in Experiment 1) a SNARC effect can emerge by using a parity judgment task. If this effect will be found, the manumerical hypothesis (*Fischer & Brugger, 2011*) can be discarded as the origin of the SNARC effect, in favor of the MNL (*Dehaene, Bossini & Giraux, 1993*). This hypothesis was tested by presented 1-to-5 magnitudes by means of both Arabic numbers and hands images, in order to also asses the possible effect of the embodied cognition on the SNARC effect.

### Material and methods
#### Participants
A subsample of 48 participants (24 female), randomly selected from the sample of Experiment 1, carried out a computerized task in Experiment 2. Participants were right-handed, with a mean handedness score of 64.67 ($\pm$ 1.79), and with an age comprised between 19 and 28 years old (age: 23.21 $\pm$ 0.29 years). Forty-six participants were right-starters and 2 participants were left-starters (all participants counted with a palm-up posture), as emerged in Experiment 1. Participants were tested in isolation, they were unaware of the purpose of the study and took part at the task as volunteers.

#### Stimuli
Stimuli were constituted by images representing the following one-digit numbers: 1, 2, 4 and 5. These four stimuli were shown both in the Arabic code and by means of photographs of hands. In particular, 8 photographs of a male right hand were created: the number 1 was depicted as a closed fist with the thumb finger lifted up; the number 2 was depicted as a closed fist with the thumb finger and the index finger lifted up; the number 4 was depicted as an open hand with all the fingers lifted up except for the thumb; and the number 5 was depicted as an open hand with all of the 5 fingers lifted up. These finger configurations are those mostly used in Italy to indicate the respective numbers, and previous evidence show no differences in the processing of magnitudes presented as the participants' typical

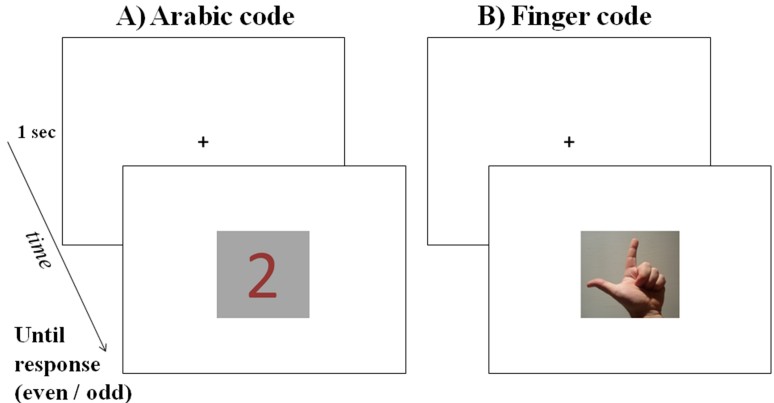

**Figure 1** Example of a trial in Experiment 2: after a fixation cross presented for 1 s, a stimulus was presented in the center of the screen until the participant categorized it as even or odd: in the figure, the magnitude 2 is presented in Arabic code (A) and in finger code (B: left-hand stimulus in palm-up posture).

finger counting configuration or in a different, culturally shared, finger configuration (*Di Luca & Pesenti, 2008*). The four finger configurations were represented both with the palm visible (palm-up) and with the knuckles visible (palm-down). The eight right hand photographs were then flipped horizontally to create a set of eight left hand stimuli. For the Arabic code, the number 1, 2, 4 and 5 were written in dark red (Calibri, font 400). All of the stimuli (Arabic numbers and hand configurations) were presented on a gray background measuring 720 × 666 pixels (height x width), corresponding to 17.76° × 16.62° of visual angle, seen at a distance of about 57 cm (screen resolution: 1,280 × 768 pixels).

### Procedure

Each participant carried out two experimental sessions composed of 200 trials each: the 16 hand photographs (8 left hand and 8 right hand) and the 4 Arabic numbers (20 stimuli) were repeated 10 times, and they were presented in a randomized order. In each trial, after the presentation of a black fixation cross in the center of the white screen for 1 s, a stimulus was presented in the center of the screen until the participant gave the response, then the subsequent trial started (see Fig. 1). Participants were asked to categorize each stimulus as even or odd by pressing two different key of the keyboard. In one session they were asked to use the left index to press "C" for even numbers and to use the right index to press "M" for odd numbers, and in the other session they were asked to use the left index to press "C" for odd numbers and to use the right index to press "M" for even numbers. The order of the two sessions was balanced among participants.

Prior to the beginning of the task, written instructions were presented and participants were informed that even numbers (2 and 4) and odd numbers (1 and 5) would be presented by means of Arabic code and finger configurations. The paradigm was controlled by means of E-Prime software (Psychology Software Tools Inc., Pittsburgh, PA), and lasted about 15 min. The experimental procedures were conducted in accordance with the guidelines of the Declaration of Helsinki.

## Results

Statistical analyses were carried out by using the Statistica 8.0 software (StatSoft, Inc., Tulsa, OK), with a significant threshold of $p < 0.05$. Due to the low numbers of left-starters, which did not allow us a statistical comparison between left-starters and right-starters, the two left-starter male participants were excluded from the analysis. Data from the 46 right-starters were analyzed by means of two analyses of variance (ANOVA). In a first step, the performance in the Arabic condition was analyzed by using Number (smaller: 1 and 2; larger: 4 and 5) and Response hand (left, right) as within-subjects factors. In a second step, the performance for the finger configuration stimuli was analyzed by using Depicted hand (left hand, right hand), Hand posture (palm-up, palm-down), Number (smaller: 1 and 2; larger: 4 and 5) and Response hand (responding hand: left, right) as within-subjects factors. In both analyses the performance obtained in the two sessions (different association between left/right hand and even/odd response) was averaged and the Inverse Efficiency Score (IES) was used as the dependent variable. IES was calculated dividing response time by the proportion of accuracy in each condition, so that a lower IES corresponds to a better performance than a higher IES, in terms of both accuracy and response time (e.g., *Prete, Malatesta & Tommasi, 2017*; *Prete et al., 2018*). In a first version of both ANOVAs, Gender of participant (female, male) was used as between-subjects factor, but it was not significant as main effect, nor in interaction with the other factors, thus it was excluded from the analyses. The mean accuracy of the sample was higher than 93% in each condition. Response times were considered only for the correct responses and when they did not exceed two standard deviations from the mean of each participant. In the ANOVAs, post-hoc comparisons were carried out by using Duncan tests.

Finally, in order to compare the performance obtained with Arabic numbers and hand configurations, a series of one sample $t$-tests was carried out, consisting in a specific $t$-test in which the statistical comparison is carried out between a distribution and a specific value used as the reference value: the performance obtained with the ''classical'' Arabic numbers was used as the reference value against which the performance obtained with hand configurations was compared. This analysis was used because we aimed at comparing the performance in the original task with hand images, to that obtained in the classical task with Arabic numbers. The results of each $t$-test specifically reveal whether there is a difference in the participants' performance between Arabic code (the reference value obtained with stimuli classically used to investigate the SNARC effect) and hand images (a set of stimuli never used before to investigate the possible embodied effect on the SNARC). Bonferroni correction was applied to correct for multiple comparisons.

In the first ANOVA (Arabic code) the main effect of Response hand was significant ($F_{(1,46)} = 5.98$, $MSE = 1595$, $p = 0.019$, $\eta_p^2 = 0.12$), showing a better performance with the right hand ($594.64 \pm 16.48$) than with the left hand ($609.04 \pm 16.79$). Importantly, the interaction between Number and Response hand was significant ($F_{(1,46)} = 96.80$, $MSE = 2539$, $p < 0.001$, $\eta_p^2 = 0.68$). Post-hoc tests confirmed a strong SNARC effect: the performance was better for larger numbers categorized with the right hand than with the left hand, and it was better for smaller numbers categorized with the left than with the right hand; left hand responses were better for smaller than larger numbers, and right hand

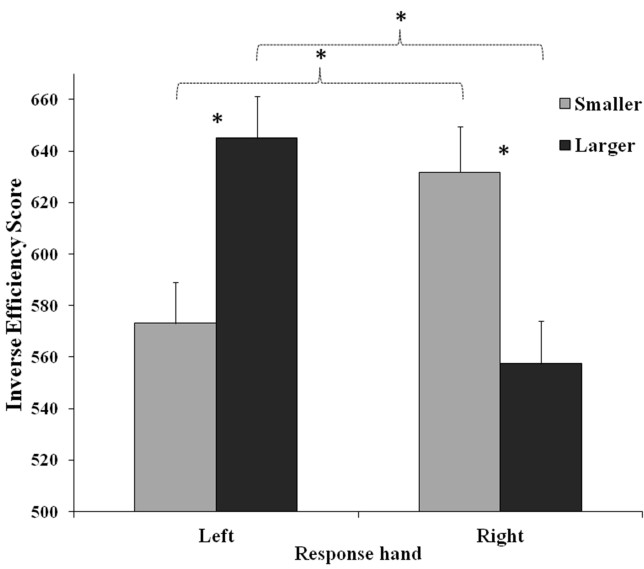

**Figure 2** **Interaction between Response hand and Number(Smaller: 1, 2; Larger: 4, 5).** The interaction is referred to the first ANOVA (Arabic code; Hand: left, right; Number: Smaller = 1 and 2, Larger = 4 and 5). Bars represent standard errors. Asterisks show the significant comparisons.

responses were better for larger than for smaller numbers (all comparisons: $p < 0.001$; Fig. 2).

In the second ANOVA (finger code) the main effect of Number was significant ($F_{(1,46)} = 22.28$, $MSE = 4766$, $p < 0.001$, $\eta_p^2 = 0.33$), revealing a better performance for larger (609.53 ± 16.95) than for smaller numbers (633.55 ± 15.47). The main effect of Hand posture was significant ($F_{(1,46)} = 5.59$, $MSE = 1450$, $p = 0.022$, $\eta_p^2 = 0.11$): the performance was better for hands depicted palm-down (618.23 ± 15.54) than palm-up (624.86 ± 16.89), confirming the effect of the embodied perspective (participants carrying out the task with their hands on the keyboard, thus in a palm-up posture). Number and Response hand significantly interacted ($F_{(1,46)} = 6.02$, $MSE = 2847$, $p = 0.018$, $\eta_p^2 = 0.12$; Fig. 3): post-hoc tests showed that the performance was better for larger than smaller numbers with both the left ($p = 0.013$) and the right hand ($p < 0.001$). Importantly, for larger numbers the performance was better with the right hand than with the left hand ($p = 0.011$), thus confirming—at least partially—the expected SNARC effect.

The significant interaction between Number and Hand posture ($F_{(1,46)} = 25.95$, $MSE = 2326$, $p < 0.001$, $\eta_p^2 = 0.37$) showed that the performance was better for larger numbers when images were palm-up than palm-down ($p = 0.027$), whereas for smaller numbers it was better when images were palm-down than palm-up ($p < 0.001$); for palm-up images the performance was better for larger than for smaller numbers ($p < 0.001$).

The interaction among Response hand, Number and Depicted hand was significant ($F_{(1,46)} = 17.17$, $MSE = 2229$, $p < 0.001$, $\eta_p^2 = 0.28$): when left hands were presented, the performance was better for larger numbers categorized with the right response hand than with the left response hand ($p = 0.002$) and for smaller numbers categorized with the left
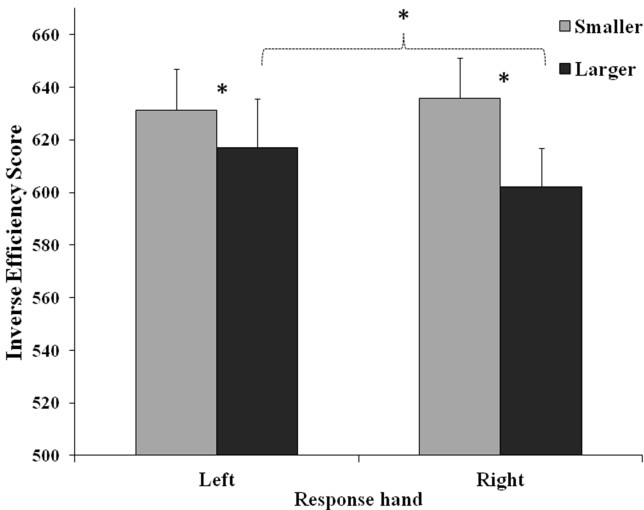

**Figure 3 Interaction between response hand and number (Smaller: 1, 2; Larger: 4, 5).** The interaction is referred to the second ANOVA (finger code; Hand: left, right; Number: Smaller = 1 and 2, Larger = 4 and 5). Bars represent standard errors. Asterisks show the significant comparisons.

than with the right hand ($p = 0.003$); moreover for these stimuli the performance was better with the right hand for larger than for smaller numbers ($p < 0.001$), whereas no significant difference was found between smaller and larger numbers categorized with the left hand ($p = 0.88$). When stimuli represented a right hand, the performance was better for smaller numbers categorized with the right response hand than with the left response hand ($p = 0.040$), and it was better for larger than for smaller numbers categorized with both the left ($p < 0.001$) and the right response hand ($p = 0.011$). Moreover, left-hand responses for smaller numbers were better when stimuli represented the left than the right hand ($p = 0.006$), whereas right-hand responses for smaller numbers were better when stimuli represented the right than the left hand ($p = 0.024$).

The interaction among Hand posture, Number and Response hand was significant ($F_{(1,45)} = 16.56$, $MSE = 2166$, $p < 0.001$, $\eta_p^2 = 0.27$): when stimuli were categorized with the right response hand, the performance was better for larger numbers shown by means of palm-up than palm-down hands ($p = 0.002$), whereas it was better for smaller numbers shown by means of palm-down than palm-up hands ($p < 0.001$). For palm-up hand images, the performance was better for larger numbers categorized with the right than with the left response hand ($p = 0.001$), and it was better for smaller numbers categorized with the left than with the right response hand ($p = 0.003$). Responses for palm-up stimuli were better for larger than for smaller numbers with both the left ($p = 0.016$) and the right response hand ($p < 0.001$).

Finally, the four-way interaction among Depicted hand, Hand posture, Number and Response hand was significant ($F_{(1,45)} = 17.89$, $MSE = 1872$, $p < 0.001$, $\eta_p^2 = 0.28$). To better understand the interaction, two ANOVAs were carried out considering palm-up stimuli and palm-down stimuli separately (see Fig. 4). Results showed that for palm-down stimuli the interaction among Depicted hand, Number and Response hand was not

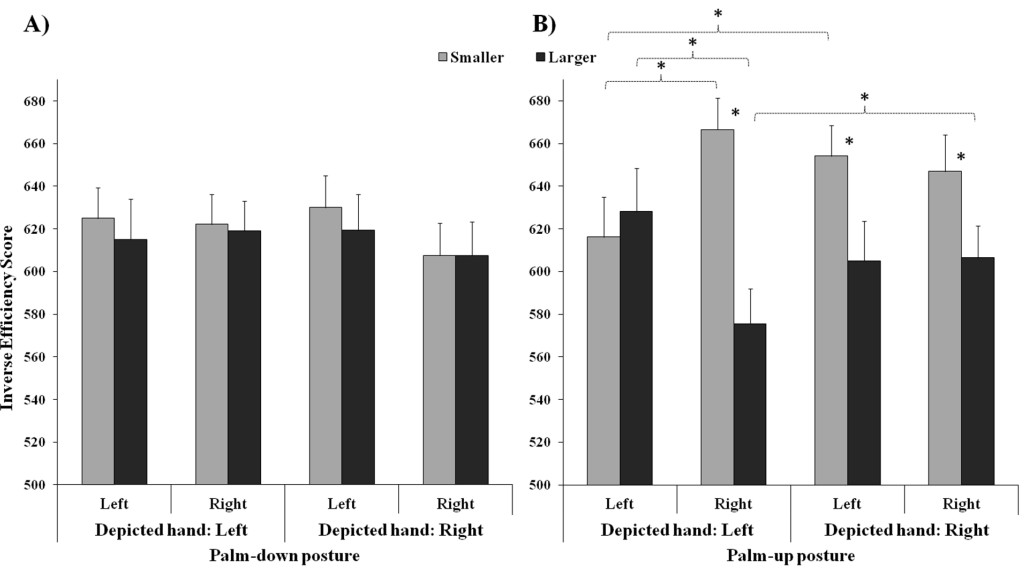

**Figure 4** Interaction among depicted hand, response hand and number: the graph shows the results obtained in two separate ANOVAs carried out for Palm-down images (A, no significant interaction) and palm-up images (B, significant interaction), separately. Bars represent standard errors. Asterisks show the significant comparisons.

significant ($F_{(1,45)} = 0.06$, $MSE = 1403$, $p = 0.812$), whereas the same interaction was significant for palm-up stimuli ($F_{(1,45)} = 26.58$, $MSE = 2698$, $p < 0.001$, $\eta_p^2 = 0.37$). Post-hoc tests showed that for left hand images the performance was better when smaller numbers were categorized with the left than with the right response hand ($p < 0.001$), and it was better when larger numbers were categorized with the right than with the left response hand ($p < 0.001$). Right-hand images were better categorized when stimuli represented larger than smaller numbers, with both the left ($p < 0.001$) and the right response hand ($p = 0.001$), whereas for left-hand images this was true only for right hand responses ($p < 0.001$). Right hand responses for larger numbers were better when stimuli represented a left than a right hand ($p = 0.009$), and left-hand responses for smaller numbers were better for left than for right-hand images ($p = 0.002$).

In order to further assess the possible difference in the processing of magnitudes presented by means of Arabic code and hand images, a series of exact $t$-tests were carried out. Specifically, IES obtained in each of the four conditions of hand images (left hand and right hand images shown palm-up and palm-down) were compared to the mean IES obtained in the Arabic code condition, for smaller/larger numbers, categorized with the left/right hand, separately (see Fig. 5). After Bonferroni correction for multiple comparisons, the results revealed that the performance was better for the Arabic code than for the hand images when larger magnitudes were categorized with the right response hand (palm-up right-hand image: $t_{(45)} = 3.36$, $p = 0.006$; palm-down right-hand image: $t_{(45)} = 3.18$, $p = 0.011$; palm-down left-hand image: $t_{(45)} = 4.46$, $p < 0.001$), and when smaller magnitudes were categorized with the left hand (i.e., SNARC effect), except for left hand images presented

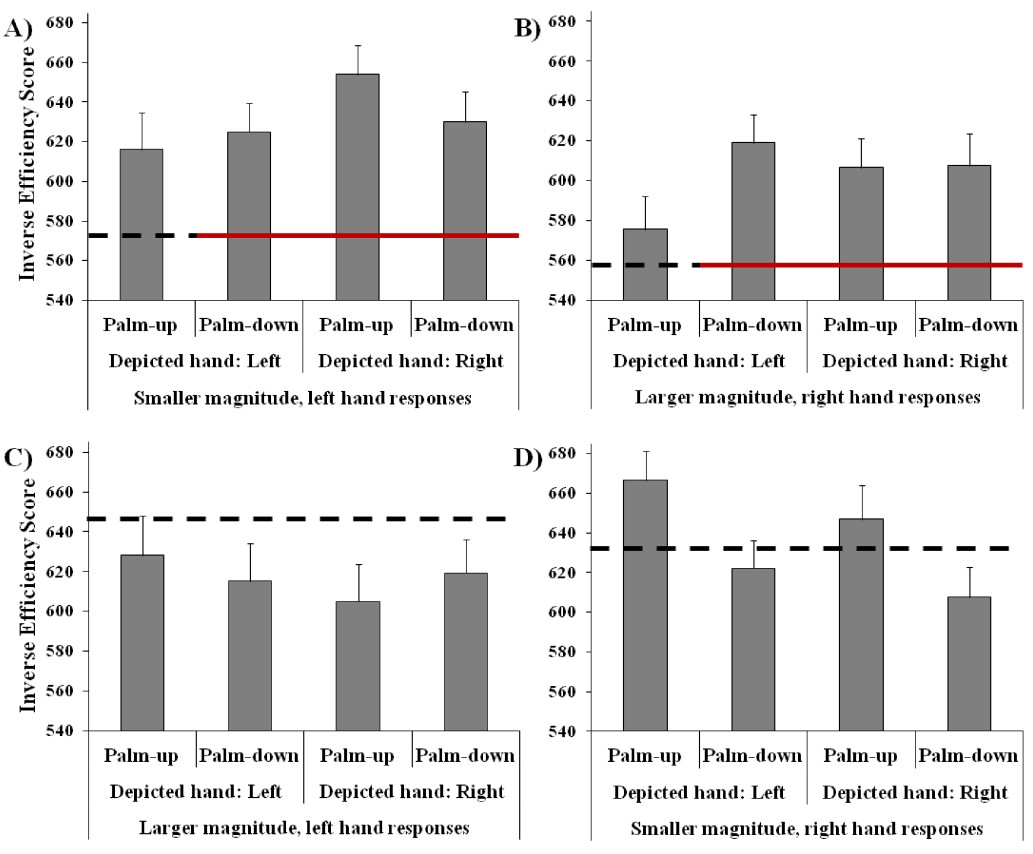

**Figure 5** Performance of participants in Experiment 2: the gray columns show the performance with hand images (bars represent standard errors). (A–B) represent the SNARC-congruent conditions (A: smaller magnitudes categorized with the left hand; B: larger magnitudes categorized with the right hand); C and D represent the SNARC-incongruent conditions (C: larger magnitudes categorized with the left hand; D: smaller magnitudes categorized with the right hand). The horizontal lines in each box represent the mean performance with Arabic code numbers (continuous line shows the significant difference, dotted line shows the non-significant difference between Arabic code and finger code).

palm-up (palm-up right-hand image: $t_{(45)} = 5.63$, $p < 0.001$; palm-down right-hand image: $t_{(45)} = 3.81$, $p = 0.002$; palm-down left-hand image: $t_{(45)} = 3.64$, $p = 0.003$). No significant difference emerged between Arabic code and hand images either for larger magnitudes categorized with the left hand, nor for smaller magnitudes categorized with the right hand (for both left and right hand images, presented palm-up and palm-down).

## DISCUSSION

The present study was aimed at assessing the finger counting direction in a Western sample (Experiment 1) and at investigating the possible relationship between finger counting and the SNARC effect (Experiment 2). Experiment 1 showed a strong population bias in finger counting direction: almost the 80% of the sample started finger counting with the right hand (right-starters). This result is in accordance with those described by *Sato & Lalain (2008)* and more recently by *Hohol et al. (2018)*. In this frame it has to

be highlighted that a left-starting bias has been reported in other studies (e.g., *Fischer, 2008*; *Lindemann, Alipour & Fischer, 2011*), by means of paper-and-pencil test and online surveys. We decided to exploit an ecological test of finger counting, starting from the evidence of the validity of this method in the investigation of the counting direction, also shown by a test-rest analysis carried out by *Hohol et al. (2018)*. We hypothesize that the different methods used to investigate finger counting (paper-and-pencil test, ecological finger counting) could be one of the reasons for the opposite patterns of results found in different studies, together with the role of the culture which seems to influence the finger counting habit at a population level (see *Lindemann, Alipour & Fischer, 2011*; *Cipora et al., 2019*). Specifically, the need to write down the direction of finger counting on a drawing depicting the two hands can be influenced by the left-to-right writing direction in the Western culture (the same holds true in the case of selecting a response from a pull-down menu). In this perspective, the left-to-right enumeration could be attributable to the habit of writing from left to right. On the other hand, the request to show how the fingers are used to count from 1 to 10 would be independent from the writing direction, and it could be more indicative of the real finger counting habits.

In Experiment 2 we confirmed the expected SNARC effect with Arabic code stimuli in a sample of participants who started finger counting with the right hand and in palm-up posture: the performance was better for smaller/larger numbers categorized (as even or odd) with the left/right hand, respectively (e.g., *Dehaene, Bossini & Giraux, 1993*; *Macnamara, Keage & Loetscher, 2018*). It should be underlined that the SNARC effect was significant, even if the magnitude range used was very small (1-to-5), showing that such a bias is really strong and it manifests itself also with a limited set of magnitudes. Moreover, we also found an overall better performance with the right than with the left hand (disregarding the magnitude), possibly attributable to the fact that all participants were right-handers.

Our results show the SNARC effect in a sample of right-starters, in contrast with the expectation based on the ''manumerical hypothesis'' (*Fischer & Brugger, 2011*). According to this hypothesis, in fact, we should expect that right-starters show a better performance with the right hand for smaller numbers and with the left hand for larger numbers, assuming that the SNARC effect is linked to the finger counting direction. Our results do not confirm this hypothesis, showing a strong SNARC effect even in a sample of right-starting participants. We can conclude that the SNARC effect is based upon a MNL instead of being dependent upon finger counting direction. Due to the fact that the present study was aimed at disentangling these two hypotheses (MNL and manumerical hypothesis), only right-starting participants were tested in Experiment 2. We found a significant SNARC effect even in a sample of right-starters, showing a relationship between finger counting and SNARC, but a direct comparison between left-starters and right-starters performance is needed in order to conclude the possible causal role of finger counting on the SNARC. In this regard it has to be highlighted that *Cipora et al. (2019)* recently found that reading/writing direction and language families have a crucial role on finger counting direction. From these premises, future studies should compare the performance of right-starters with that of left-starters, and cross-cultural studies should be carried out,

in order to further investigate the interaction between finger counting direction and culture on the SNARC effect.

Starting from the embodied cognition theory, we also aimed at exploring whether the SNARC effect is present when making numerical judgments on hand configurations. To assess this, we presented photographs of left and right hands in palm-up and palm-down postures, showing 1-to-5 magnitudes with fingers. It has to be highlighted that *Di Luca & Pesenti (2008)* showed the absence of differences in the processing of magnitudes presented by means of finger configurations, when these magnitudes were presented either as the participants' typical finger counting configuration (e.g., number four: all fingers extended except the little finger which is closed on the palm) or as "finger-montring" (the most shared finger configuration used in a culture to refer to a specific number; e.g., number four: all fingers extended except the thumb which is closed on the palm). The authors concluded that both of these finger configurations are stored in long-term memory and are associated with the respective magnitudes. Starting from this evidence, in Experiment 2 we used the "finger-montring" configurations, even if they could be different from the finger counting configurations found in Experiment 1.

Our results confirmed the SNARC effect also when stimuli were hand images, at least partially: the performance was better for larger magnitudes categorized with the right than with the left hand, confirming the independence of the SNARC effect from a specific code. This evidence is in accordance with previous results (*Nuerk, Wood & Willmes, 2005*) showing that the SNARC effect withstands to different magnitude notations (Arabic numeral, visual number word, visual dice pattern), as well as to different perceptual modalities (visual and auditory number words). Moreover, we also explored the possible effect of the hand images presented as stimuli (left hand, right hand) as well as the effect of the hand posture (palm-up and palm-down), and we found a better performance for smaller/larger magnitude categorized with the left/right hand (SNARC effect) when stimuli were left hand images. We also found that in palm-up condition, the performance of the participants was better for left hand images than for right hand images when smaller/larger magnitudes were categorized with the left/right hand, respectively. These results seem to suggest that a SNARC effect can be found with hand images, mainly when stimuli represent left hands, in palm-up posture, suggesting a kind of interaction between the MNL and hand representation: we could speculate that also these results can be intended as linked to the MNL, in which all of the magnitudes presented here (from 1 to 5) could be placed on the leftmost portion of the line, possibly being considered as "small" numbers, thus privileging a left hand representation. Accordingly, the results of the $t$-tests further confirm that the SNARC effect is stronger for magnitudes presented in the Arabic code than by means of hand images: the performance of participants was better in categorizing magnitudes in Arabic than in finger code in the conditions congruent with the SNARC effect (smaller magnitudes categorized with the left hand and larger magnitudes categorized with the right hand) and they did not differ in the conditions incongruent with the SNARC effect (smaller magnitudes categorized with the right hand and larger magnitudes categorized with the left hand). These significant differences were present for right hand images shown palm-up and palm-down, as well as for left hand images shown palm-down. The difference

between the two codes was not significant for palm-up left hand images, confirming that the task benefits from the same facilitation (SNARC) when stimuli are in Arabic code and left hand palm-up configurations. In other words, we found that the parity judgment task required the same cognitive effort when magnitudes were presented either in Arabic code or by means of hand configurations, in the conditions incongruent with the SNARC effect. Nevertheless, in the SNARC-congruent conditions the performance was better for Arabic code than for hand images, confirming that the Arabic code is the preferential code for magnitude processing, except for magnitudes presented by means of left hand palm-up images. This pattern of results shows that left hand palm-up images facilitate the magnitude processing in the same way as the Arabic code, in accordance with the spatial numerical association, and we speculate that this effect could be linked to a possible interaction between the SNARC effect and the left-to-right MNL. In this frame, the MNL could explain a left-hand preference for 1-to-5 magnitudes, even if it does not correspond to the real finger counting habit. However caution is needed in this regard, due to the evidence that the magnitude of the stimuli presented seems to be mentally categorized as small or large in accordance with the specific range of magnitudes used (*Dehaene, Bossini & Giraux, 1993*).

For smaller magnitudes we also found a hand compatibility effect: left-handed responses were better for left than right hand images, and right-handed responses were better for right than left hand images. This pattern of results confirms a body-related processing, for which stimuli representing one hand are better categorized with the corresponding response hand. The fact that the performance of participants was overall better when hand images were shown as palm-down can be associated to the specific hand postures of the participants during the task (with the finger on a keyboard, i.e., palm-down). In this perspective we can conclude that embodied cognition extends to hands representation and it can modulate the responses of participants in a task not requiring an explicit representation of the body posture (see *Di Luca et al., 2006*; *Riello & Rusconi, 2011*). However, the interaction revealing that the palm-down configuration enhanced the performance only for larger magnitudes (smaller magnitudes being better processed when presented palm-up), prevent us to support this hypothesis conclusively.

We also found that when stimuli were hand images, the performance was better for larger than for smaller magnitudes, with both responding hands, an evidence not found in the Arabic code. This evidence shows that different codes are processed in different ways, and future studies are needed to verify which kinds of codes allow magnitude representations to interact with the MNL. Finally, no effect of participant gender was found with both Arabic code and hands images, suggesting that females and males do not differ in this task. Since hands images were created from a male hand photograph one could expect a better performance by male than female participants, due to an own-gender bias (e.g., *Prete et al., 2016*), but the absence of significant differences between female and male participants seems to disconfirm such a bias in the parity judgment task. Nevertheless, future studies in which both female and male hands photographs are presented will definitively disentangle the possible gender difference in this domain.

## CONCLUSIONS

We can conclude that the right-to-left hand counting direction, widely found in an Italian sample by means of an ecological task, does not prevent the occurrence of a SNARC effect: this expected effect was confirmed in a sample who started counting by using the right hand. This is the most crucial result of the present study, supporting that finger counting habits and the mental number line are different processes. The SNARC effect is present not only for Arabic code numerical stimuli, but it can be detected also when a different code is used, such as finger configurations corresponding to 1-to-5 magnitudes. Finally, our results also suggest that all of these effects could be interpreted in the light of embodied cognition, but—due to the multiple interactions found—further studies are needed to disentangle to which extent body representations can influence numerical cognition.

### Funding

The authors received no funding for this work.

### Competing Interests

Luca Tommasi is an Academic Editor for PeerJ.

### Author Contributions

- Giulia Prete conceived and designed the experiments, performed the experiments, analyzed the data, prepared figures and/or tables, authored or reviewed drafts of the paper, and approved the final draft.
- Luca Tommasi conceived and designed the experiments, authored or reviewed drafts of the paper, and approved the final draft.

### Human Ethics

The following information was supplied relating to ethical approvals (i.e., approving body and any reference numbers):

The present study did not involve patients, children or animals, as well as drugs, genetic samples or invasive techniques, thus it was not subject to ethical review by the academic medical research board. Nevertheless, verbal informed consent was obtained from all participants and the experiment was conducted in accordance with the ethical standards prescribed by the Declaration of Helsinki.

### Data Availability

Raw data are available in the Supplemental Files.

### Supplemental Information

Supplemental information for this article can be found online at http://dx.doi.org/10.7717/peerj.9155#supplemental-information.

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
