# Peer review of "Exploring the interactions among SNARC effect, finger counting direction and embodied cognition"

_PeerJ, doi:10.7717/peerj.9155_

## Round 0.1 · original submission · Major Revisions

I have received two reviews on your paper from experts in the field. Both reviewers express a major revision verdict, and I share their opinion.
The reviews are clear and constructive, so please refer to them for details. The theoretical part, and the reporting and interpreting of the results could be ameliorated. Furthermore, I agree with Reviewer 1 that you would provide a more convincing case if you tested left-handed.

·

Basic reporting

No comment.

Experimental design

As I outline in the comments to the author, the design as it is now is not suited for addressing the experimental question stated by the authors.
Simply - in the light of literature it is not possible to investigate influence of finger counting starting hand on the SNARC effect if only participants starting from their right hand are tested. It is not a variable but a constant then. Moreover, in the literature there are multiple hints of evidence that right-starters do reveal SNARC effect, but it seems that their SNARC is weaker than the SNARC by left-starters. However, testing right starters only does not address the issue.

Validity of the findings

As outlined above and in more detail in the general comments - as for now (no left-starters tested with the SNARC task), the results are hardly conclusive.

Additional comments

First of all, I must say that I served as a reviewer to this manuscript few months ago, when it was submitted to another journal, and rejected after a peer review by the editor. I paste this review below. In the submitted version the authors did not consider any of these points. I still believe these need to be addressed and hardly believe that it is possible to address them without additional experiment (testing left-starters with the SNARC task). However, if the Authors provide good arguments, I can reconsider this opinion.

Below I paste my old review.
* * *
n the paper the authors investigate relationship between SNARC and finger counting routines. In particular, they investigate on how right-to-left finger counting routine relates to the SNARC, trying to disentangle between the Mental Number Line account and the “Manumerical” hypothesis. The research question stated by the authors is valid and theoretically sound. Nevertheless, I have numerous comments and doubts, which I list below. I think they can be addressed within a major revision. However, I am not sure, whether it is possible to handle these comments without additional experiment. If the authors come up with good arguments, that it is not necessary, I can reconsider my viewpoint.

My biggest concern regarding the study spreads from literature coverage and extends to approach taken for data collection, and its interpretation, so I will try to describe my point according to this order.

One of the basic arguments raised by the authors is that methodological differences have led to discrepant results as regards the most prevalent finger counting routine. I generally agree that this can be a point. Actually the authors should also mention that this was empirically demonstrated by Wasner and colleagues (2014, Cognitive Processing). On the other hand, despite obvious methodological differences, I believe there are still some cultural (even within the cultures using the same reading/writing direction) differences in finger counting routines (as the authors themselves report studies from numerous European countries). One might argue that these differences can be due to methodological differences. However, this is not the whole story. There are at least two large-scale studies showing that using the same (even if not ecologically valid) procedure, reveals massive cultural differences in reported finger counting routines. This gets pretty obvious e.g., when one inspects Figure 2 in Lindemann et al. (2014). Proportion of left-starters differs from ca. 85% in the US participants to about 50% in Belgian and Italian participants. Cipora et al (2019, Behavior Research Methods; Supplementary Material, Part 5) also report massive between country differences in finger counting routines. I can hardly understand on how the method validity (if this was the only case) leads to different biases in different cultures.

Another related point here. If the authors wish to discuss the minor methodological differences between studies, which might have led to inconsistencies in results, the review of existing evidence must be done at a more fine-grained level regarding differences between methods. For instance, in study by Hohol and colleagues (2018) participants did not see the questionnaire while asked to count with their fingers, but were asked count with their fingers first, and then were given a questionnaire and asked to mark their responses (this is actually indirectly mentioned in the discussion when the authors write about the validity of the ecological method, but in the introduction it only states that modified version of Fischer’s questionnaire was used). This could differ from a situation if the drawings are constantly visible and participants are only to mark their responses without actual counting.

Here I come to another important point, which refers to the approach taken by the authors. Specifically, only right starters have been tested in Experiment 2 (except two left-starters, who were excluded from the analysis anyway). Testing right-starters only stands in contradiction to the opening paragraph of the Discussion section where the authors argue about independence of the SNARC from finger counting. How this conclusion can be stated if only one group is tested? To test it, right-starters should be compared with left-starters.

The observation by Fischer (2008) that there is no SNARC in right-starters seems not to be a strong enough argument to make investigating only right-starters legitimate. Robust group level SNARC was observed multiple times in cultures, in which vast proportion of individuals are right-starters.

In a recent study Cipora and colleagues (2019; Behavior Research Methods; Supplementary Material Part 2) replicated the difference between right- and left-starters in the SNARC. Nevertheless, despite group difference (effect size was actually very small), the robust SNARC was found in both groups.

In the current study it is impossible to verify it as only right-starters have been tested, and it is hard for me to find (nor think of) any solid conclusion that can be drawn from the study as it is now. For the same reason, I am also not sure whether the results reported by the authors make it possible to draw such a strong conclusions as regards the nature of the SNARC, as the authors do on upper part of page 19.

Other points:

Introduction:

This section would benefit from streamlining. For quite a long time I cannot follow the logic of the literature review presented by the authors as regards the goals of this literature review.

Maybe information on failure to replicate the crossed hands SNARC should be put next to information on observing the effect. This can be more accessible to the readers. In the context of hand and their position for the SNARC raised in this paper, it would also be worthwhile to mention a study by Viarouge et al. 2014 (QJEP).

Description of Fischer (2008) study is a bit misleading because provided sample size and percentage comes from his Experiment 1, which only showed prevalence of the right- and left-starters. SNARC was evaluated in a subsample of this in Experiment 2.

Hypotheses and Method:

The authors build rather elaborate prediction about hand position and its correspondence to actual position of hands of the participants during the experiment. They also do emphasize that they wished to have the hand images as natural as possible. As in principle I see a point in that approach, I have one doubt. Only photos of male hands were used. From studies on immersion in virtual reality one can infer that similarity between own body and the body displayed in VR environment plays a considerable role for the immersion experience. For this reason one could expect different effects for males and females in the study. This issue should be at least mentioned and discussed by the authors.

Results:

A short paragraph / sentence summarizing conclusions of Experiment 1 could be added before moving to Experiment 2.

Experiment 2 – in the part on number presented as hand pictures, the authors are investigating a four-way interaction. The description is fairly hard to follow (see Halford et al., 2005, Psych Science for an evidence that humans have considerable difficulties to mentally process such complex interactions). I would consider streamlining the description and making sure this is accessible to the reader. Maybe a table summary would help? I have no clear solution to that.

What do you mean by exact t-tests? Does it refer to follow-up (post hoc) analyses, or a very specific method in multivariate statistics (e.g., Lauter, 1996)? If it is the latter, please provide a short footnote explaining it, because it is not a commonly used method.

Discussion:

This section would also benefit from streamlining, and in principle I think that multiple changes need to be implemented as a consequence of considering the above arguments.

p 18 last part is framed as a hypothesis that the measurement method influences the results of finger counting questionnaire. I think this is not a good framing here, the authors have actually not tested that.

Reviewer 2 ·

Basic reporting

### Language:
Although I am not a native English speaker myself, I noticed several odd sentences and grammatical errors in the manuscript. The manuscript would profit from proof reading by a native English speaker.
In parts (especially the last section in the introduction (hypotheses exp 2) and the results section of experiment 2) the manuscript is not clearly written. It would be helpful to more explicitly/transparently describe the train of thoughts/reasoning behind certain statements or analyses. Especially in the results section, it is useful to reiterate the hypothesis that is being tested with a specific analysis/mention whether or not the expectations were met when presenting the results. Or – vice versa – describe the expected effect (main effect, interaction) more explicitely in the hypotheses section.

### Literature
The authors give a broad overview of relevant literature. Although the authors present a multitude of studies relating to the one or the other research question, a line of arguments that leads to the respective research questions is hard to filter out for the reader. Put differently, the presentation of previous research is not well structured and the relevance to the study presented is not always clear. The authors should try to make one argument per paragraph (maybe working with subheadings if it helps following the line of arguments) and end a paragraph with the important statement/conclusion of this paragraph for the study at hand. As one example, the in lines 97-152 (that is more than two pages) the authors mix (amongst other things) research on the manumerical hypothesis and findings regarding effects of situatedness on counting directions.


### Figures/Tables
Quality of Figure 1 might have to be checked

### Data
Aggregated data of experiment 2 has been made available (I guess the numbers are inverse efficiency scores but this is not specified in the file). Raw data of experiment 2 are not provided.

Experimental design

### Research questions
The presented study addresses multiple research questions, of which I think the first two are described clearly in the hypotheses section (1. Assess the proportion of right- and left-starters in an Italian sample, 2. SNARC effect with a restricted stimulus set (Arabic digits)). After this I have a hard time grasping the research question and especially also the justification of hypotheses.
1. “… and to verify the congruency between the finger counting direction (i.e., left hand use for small numbers) and the SNARC effect (i.e., better performance with the left hand for smaller numbers).” I do not understand what the authors are referring to here because in Experiment 2 only right starters were investigated so I don’t see how the authors want to investigate the association of counting direction and the SNARC effect (neither with Arabic digits or hand pictures).

2. “We expected that, according to the SNARC effect, for both codes the performance of participants would be better for smaller/larger numbers categorized with the left/right hand, respectively.” I assume that “code” refers to Arabic digit and hand stimuli (this is not completely clear to me because on the lengthy description that proceeds this hypothesis). If so, the interaction of number x response hand interaction should be present for both Arabic digits and hand stimuli (?)

3. “We also hypothesized that, according to an embodied perspective, the photographs of a left/right hand would lead to a better performance with the corresponding left/right hand used to respond.” Interaction of depicted hand and response hand (?)

4. “Finally, we were also aimed at exploring whether this hand-compatibility effect could be stronger for stimuli presented palm-down (as this was the posture of the participants’ hands during the task) or it was stronger for stimuli presented palm-up (as the viewed hands during a first-person finger counting). Thus our hypothesis was that of a better performance with the left/right hand for smaller/larger magnitudes (SNARC effect) shown by means of left/right hand images (embodied cognition) presented either palm-up (as in first person finger counting) or palm-down (stronger effect of the embodied cognition than perceptual habit).” First, it is unclear if this is an exploratory or a confirmatory analysis. Nevertheless, the hypothesis the authors formulate here is unclear to me.

Validity of the findings

### Discussion
1. The authors state that “Our results show that the SNARC can be independent of the finger counting habits.” (line 468f), and that “We can conclude that the SNARC effect is based upon a MNL instead of being dependent upon finger counting direction.” (line 473f, see also conclusions drawn in the conclusion section and statement in the abstract). Neither statement is justified because only right-starters were tested. To be able to judge potential influences of counting direction, counting direction needs to be manipulated.

2. Following from the fact that the research questions in experiment 2 are not clearly derived, both the reporting of results and the discussion of these results do not provide a clear picture of what is going on and what it might mean. In the results section, a series of results is reported, partly reiterated in the discussion section and the general message is hard to grasp. I think it is necessary to focus the analyses, tailor them to clearly formulated questions that address a specific gap. Otherwise it remains a list of many results lacking a clear contribution to existing research or prompting further studies.

### Conclusions
Given that a SNARC effect only seems to be present under very specific circumstances with hand stimuli (and was found in an exploratory analysis as I understand) arguing that “The SNARC effect is a pervasive effect which is present not only for Arabic code numerical stimuli, but it is also present when a different code is used, such as finger configurations corresponding to 1-to-5 magnitudes.“ is an overstatement.

Additional comments

### Other comments
- The authors should check the use of “frame of reference” (line 171 This frame of reference, line 183 object-base frame of reference, line 184 egocentric frame of reference etc.).
- Analysis counting direction and handedness score: This analysis wasn’t introduced in the introduction and is not discussed in the discussion. Also, why make three groups, if there are only 5 people in one group/ why perform an analysis on 5 people?
- Which post hoc test was used? (Duncan, line 334, or Bonferroni, line 427)
- E.g., line 233 (s/he): The singular “they” is a generic third-person singular pronoun in English. Use of the singular “they” is endorsed as part of APA Style because it is inclusive of all people and helps writers avoid making assumptions about gender.

---

## Round 0.2 · Minor Revisions

I have now received one review from the original reviewer of your paper. The reason of the delay in getting back to you is that I was waiting for the further reviewer, but we decided to take a decision due avoid waiting any further. As you will see, Reviewer 1 recognizes that the manuscript is improved, and proposes some minor revisions. I invite you to revise the manuscript following his/her suggestions.

·

Basic reporting

No comment here, but see comments in the section on validity of findings. These comments refer also to this point, but it was really hard to split them.

Experimental design

No comment here, but see comments in the section on validity of findings. These comments refer also to this point, but it was really hard to split them.

Validity of the findings

While reading the revision and responses to the reviews, I think that making it explicit that the right starters are the target sample somehow helps defending the paper as it is now. However, I have a feeling that the “Manumerical hypothesis” as framed by the authors seems to be a little bit a strawman. Specifically, as framed by the authors, the “manumerical hypothesis” claims that finger counting is THE ONLY factor determining the SNARC (not sure whether such understanding will be even shared by M. Fischer, who proposed it). In that case the experiment allows verifying it. As a side note, even in original Fischer’s study, the SNARC in right starters would be significant if tested one-sided (note that one sided tests are actually recommended by several authors in the case of the SNARC effect because there is a clear directional prediction). Moreover, the null result as reported in the paper was probably also due to presence of two participants revealing very positive slopes (> 40).
On the other hand, what the current design does not address is whether the finger counting habits INFLUENCE the SNARC. It can well be, that right-starters reveal weaker SNARC than left-starters. This is what we have actually found in the online study, I mentioned. Please note that, according to Fig 2 in Lindemann (2011, not 2014, sorry for a typo in the previous review), in Italy and Belgium the proportions of left-starters is only about 50%, while multiple SNARC studies come from these countries. Does this mean that the “manumerical hypothesis” holds or not? I think this remains an open question.

Putting aside my personal ideas on which version of the hypothesis is more interesting, I believe, verifying the “bold” version of “manumerical hypothesis” has some value as it is. Therefore, this manuscript provides a nice systematization of evidence present here and there in the literature and adds up some new and interesting data collected with the aim of testing this particular hypothesis. However, it must be clearly stated what the results tell (“bold” version does not hold) and what they do not because of not testing a group of left-starters (i.e., they do not say whether finger counting routines play a role”).

The conclusion that future studies should compare left- and right-starters: this is actually what was done in the online study in a quite large scale. I personally support replication efforts as a way to address several problems of psychology, but the framing as proposed by the authors states that it would be a kind of novel way to pursue the question, which is actually not the case.

Additional comments

Abstract (line 33): there is still a confusing sentence on independence of the SNARC from finger counting habits. Please adjust in line with adjustments in the main text.

Lines 54-55: the description of the SNARC effect is somehow weird. It is a mixture of presenting the effect and (probably) reporting of Experiment 3 from Dehaene et al., 1993. As it is now: “were faster at categorizing smaller numbers (from 0 to 5) with the left hand and larger numbers (from 4 to 9) with the right hand”, the sentence contradicts itself – numbers 4 and 5 are claimed to be responded faster by left and right hand at the same time. I know what you probably meant here but the formulation is not very fortunate.

Section 1.2 – I still found it hard to follow. Would benefit from streamlining and making it clear on why certain information / arguments are provided. Not fully sure about the relevance of the neuro data. It should be more integrated into line of argumentation or skipped.
Not sure why the MARC effect is introduced at this moment and how does it relate to previous argument.
I am not sure what the authors wanted to convey in the first paragraph of this section and how it was meant to relate to the second paragraph.

I am not fully sure whether citation of a work by Schroeder and colleagues (2019) is relevant here. Obviously, being involved in this paper I have no interest in kicking out this citation (rather the contrary), but please mind that this publication is stage 1 registered report, where we outline some theory about the MARC effect justifying the study to be conducted, but no data has been collected yet. If you referred to the theoretical consideration, consider making it explicit.

Line 138. The added text is misleading because it is meant to refer to study by Linemann and colleagues (I guess) but there is no citation and the previously cited work is Wasner and colleagues. As far as I remember, the latter did not consider cross-cultural comparisons (correct me if I am wrong).

“Exact t-test” – I am still not convinced that one-sample t-test is also known as “exact t-test”. Quick googling also did not solve it. Why not sticking to one-sample t-test as it is known in most statistical packages and handbooks? The term one-sample t-test is also quite well-known to SNARC researchers.
Related point – why not comparing two conditions directly but using a mean in one condition as a reference? An analysis as proposed by the authors would make sense to me if: (a) comparison is made to a specific value (e.g., zero), which would have a certain theoretical meaning, (b) one tests a specific group / sample (such as patients or so) against well-established population norm population norm. Why doing it here?
If I am missing the point completely, please correct me, and consider explaining it in simpler words. I could not follow how many such t-tests were run and what was actually compared to what.

---

## Round 0.3 · accepted · Accept

I am happy to inform you that your paper has been accepted for publication on PeerJ.